Effects of home-based telerehabilitation on dynamic alterations in regional intrinsic neural activity and degree centrality in stroke patients

Chen Jing 16111290003@fudan.edu.cn 1
Li Jing 1
Qiao Fenglei 2
Shi Zhang 3
Lu Weiwei 4
1 Department of Neurology, Zhongshan Hospital, Fudan University , Shanghai , China
2 Department of Rehabilitation, The Shanghai Fifth People’s Hospital, Fudan University , Shanghai , China
3 Department of Radiology, Zhongshan Hospital, Fudan University , Shanghai , China
4 Department of Rehabilitation, Zhongshan Hospital, Fudan University , Shanghai , China
Liu Feng
Electronic publication date: 2023 Sep 1
Publication date: 2023
Volume: 11
Electronic Location ID: e15903
Received 2023 Apr 9; Accepted 2023 Jul 25
Copyright: ©2023 Chen et al.
Copyright year: 2023
Copyright holder: Chen et al.
License: This is an open access article distributed under the terms of the Creative Commons Attribution License, which permits unrestricted use, distribution, reproduction and adaptation in any medium and for any purpose provided that it is properly attributed. For attribution, the original author(s), title, publication source (PeerJ) and either DOI or URL of the article must be cited.
License URL: https://creativecommons.org/licenses/by/4.0/

Keywords: Telerehabilitation, Dynamic intrinsic brain activity, Regional homogeneity, Amplitude of low-frequency fluctuations, Degree centrality

Funding: National Natural Science Foundation of China 82101391 The study was supported by the National Natural Science Foundation of China (82101391). The funders had no role in study design, data collection and analysis, decision to publish, or preparation of the manuscript.

==============================
Objective

To explore the effects of home-based telerehabilitation (TR) on dynamic alterations in regional intrinsic neural activity and degree centrality in stroke patients by resting-state functional MRI (fMRI) methods.

Methods

The neuroimaging data of 52 stroke patients were analyzed. Dynamic regional spontaneous neural activity (dynamic amplitude of low-frequency fluctuations, dALFF; and dynamic regional homogeneity, dReHo) and dynamic degree centrality (dDC) were compared between the TR and conventional rehabilitation (CR) groups. A flexible factorial model was employed to investigate the expected effects.

Results

The patients in the TR group showed increased dALFF in the right precuneus and bilateral precentral gyrus (PreCG) and reduced dALFF in the right inferior parietal lobule by the analysis of main effects. Significant differences between groups were detected in the right precuneus, right fusiform gyrus and left middle frontal gyrus for dReHo and in the left cingulate gyrus, right middle temporal gyrus and left precuneus for dDC. A significant correlation was found in the TR group between the changed dALFF in the left PreCG and the changed Fugl-Meyer assessment (FMA) scores from baseline to postrehabilitation.

Conclusions

This study implied that home-based TR training can alter the patterns of dynamic spontaneous brain activity and functional connectivity in certain brain regions. The identification of key brain regions by neuroimaging indicators such as dynamic regional brain activity and degree centrality in the recovery process would provide a theoretical basis for noninvasive brain stimulation technology and strategies for formulating targeted rehabilitation programs for stroke patients with motor dysfunction.

Introduction

Stroke is among the most common causes of acquired disability among adults in China (Zhang et al., 2022). It has been reported that motor functional disorders are the most prominent symptom after stroke and leads to long-term restrictions in activities of daily living (ADL) and long-term costs for stroke patients (Sennfalt et al., 2018; Ma et al., 2021). Evidence has shown that rehabilitation is the most effective means for addressing motor disability after stroke (Stein et al., 2021; Asakawa et al., 2017). However, the proportion of patients with stroke who access rehabilitation is very low because of the relative shortage of hospital rehabilitation resources in China (Asakawa et al., 2017). In addition, in particular periods, such as during outbreak and prevalence of COVID-19, stroke patients had to stay at home to cut off the transmission route; consequently, patients lost the opportunity to receive rehabilitation training in hospitals during the golden period of rehabilitation after stroke. Thus, home-based rehabilitation approaches need to be emphasized and developed.

A previous randomized controlled trial investigating the effects of a home-based telerehabilitation (TR) approach on stroke patients demonstrated that a telerehabilitation program significantly improved motor function recovery and enhanced resting-state functional connectivity between the bilateral precentral gyrus (PreCG) compared with conventional rehabilitation (CR) (Chen et al., 2020a; Chen et al., 2020b). Moreover, a previous study demonstrated that the functional reorganization of interhemispheric PreCG areas was significantly associated with motor function improvements (Chen et al., 2020a; Chen et al., 2020b).

However, the discrepancies in regional brain activities and brain core-hub areas of functional change in stroke patients receiving TR or CR training are still unclear, which is not conducive to understanding the rehabilitation mechanisms. Neural regional properties are critical in helping us better understand the neuro-physiological and neuro-pathological situations. The amplitude of low-frequency fluctuations (ALFF) tests the signal strength in low-frequency oscillations of spontaneous brain activity and reflects regional spontaneous synchronous neural activity (Zang et al., 2007). Regional homogeneity (ReHo) assesses resting-state regional neural spontaneous activity, reflecting the statistical similarity of regional brain activity among adjacent areas in space (Zang et al., 2004). Hence, these two metrics have been used to evaluate regional brain function and the efficiency of rehabilitation therapies in neurologic disorders (Liu et al., 2013; Liu et al., 2014). Degree centrality (DC) is a graph theory-based measure of how strongly a certain region is connected to other regions, has been applied to evaluate alterations in nodal importance in brain functional changes in neurologic disorders and neuropsychiatric diseases (Zuo et al., 2012; Wang et al., 2019; Chang et al., 2022).

As spontaneous neural activity in the brain is characterized by dynamic changes over time and the brain network always attempts to keep maintain a dynamic balance of function in constant change, static approaches may ignore the dynamic features of local neural activity and brain functional connectivity (Liu et al., 2017a; Liu et al., 2017b; Xue et al., 2022). Investigating the dynamic features of intrinsic local brain activity and functional connectivity over time may reveal the adaptive modulation of neural function for rehabilitative therapeutic response in stroke patients. The joint use of dynamic rather than static indices of regional intrinsic neural activity and brain functional connectivity made it easier to acquire more objective and realistic information.

Studies have reported that the model of spontaneous neural activity variability has been changed in patients with motor deficits after stroke, and dynamic indices of regional intrinsic neural activity might be potential methods to measure motor function (Chen et al., 2018a; Chen et al., 2018b; Tian et al., 2022). Moreover, an increasing number of studies have used static neuroimaging indices to evaluate the effects of rehabilitation on motor function after stroke and to explore the potential rehabilitation mechanism of brain-computer interface assistance (BCI), transcranial direct current stimulation (tDCS), repetitive transcranial magnetic stimulation (rTMS), intensive occupational therapy and other rehabilitation therapies (Hu et al., 2021; Cui et al., 2022; Wanni et al., 2021). However, no studies have investigated the dynamic characteristics of spontaneous neural activity and functional connectivity in stroke patients with motor deficits during the process of TR. Given the previous results that TR tends to promote motor function recovery and has different brain functional reorganization patterns compared with CR (Chen et al., 2020a; Chen et al., 2020b; Chen et al., 2017), in the current study, we hypothesize that home-based TR training would manifest different dynamic change patterns in intrinsic brain activity and DC variability compared with CR training and that the changed dynamic regional indices or DC map in certain brain areas would be observed to be related to the improvement of motor function. To explore this possibility, the data from a previous clinical trial were reanalyzed to determine the effectiveness of home-based TR with respect to the dynamic alterations in regional intrinsic neural activity and degree centrality in patients with stroke and to investigate the relationships between the changed dynamic regional indices or DC map and Fugl-Meyer assessment (FMA).

Methods

Study design and patients

This study used existing data from a randomized, single-blind, controlled parallel group study in which home-based TR training or CR training was given for 12 weeks to 52 subcortical stroke patients in China (Chen et al., 2020a; Chen et al., 2020b). See Supplemental Files for patient exclusion and inclusion criteria. This study was registered with the Chinese Clinical Trial Registry as ChiCTR-IPR-17011757 and was conducted in compliance with the protocol (Chen et al., 2018a; Chen et al., 2018b). Written informed consent was obtained from all subjects. The details of the study were described in a previous report published in 2020 (Chen et al., 2020a; Chen et al., 2020b).

Randomization and blinding

Participants who met the eligibility criteria were randomly assigned in a 1:1 ratio to the TR group or CR group. Given the nature of the rehabilitation intervention, it was impossible to blind the patients, caregivers, and therapists about allocation and intervention; only outcome assessors, MRI data acquisition staff, and data analysts were blinded.

Intervention

Intervention measures were described in detail in previous studies. Patients either in the TR group or CR group were required to receive rehabilitative intervention, including occupational therapy (OT), physical therapy (PT) and electromyography-triggered neuromuscular stimulation (ETNS). The amounts of OT/PT and ETNS were registered for each patient by either the patients or the caregivers (in the TR group) and the therapists (in the CR group). Included patients were assessed at baseline (T1), after the rehabilitative interventions at the week 12 (T2) and at the end of the follow-up period at the week 24 (T3).

Outcomes

Dynamic alterations in regional intrinsic neural activity were the dynamic ALFF (dALFF) and dynamic ReHo (dReHo). Dynamic DC (dDC) was used to measure dynamic alterations in functional connectivity. FMA scores were evaluated as motor function for the upper and lower extremities.

MRI data acquisition

The MRI data were acquired via a 3.0T Philips Achieva MR scanner (Philips Medical Systems, Best, Netherlands) with a 32-channel head coil. Tight but comfortable foam padding and earplugs were applied to minimize head motion and to reduce scanner noise. During scanning, all participants were instructed to remain awake, keep their eyes closed, stay motionless and attempt to think of nothing. The imaging protocols included the following parameters: (1) resting-state functional MRI (fMRI) was scanned using an echo-planar imaging (EPI) sequence: repetition echo time (TE) = 30 ms, repetition time (TR) = 2000 ms, field of view (FOV) = 220 mm × 220 mm, flip angle (FA) = 90°, matrix = 64 × 64, slice thickness = three mm, gap = one mm, voxel size = 3 mm × 3 mm × 3 mm, and 180 volumes; (2) high-resolution sagittal T1-weighted images were obtained by a three-dimensional magnetization-prepared rapid gradient echo (MP-RAGE) sequence: TE = 3.7 ms, TR = 8.0 ms, FOV = 256 mm × 256 mm, FA = 12°, matrix = 256 × 256, slice thickness = one mm, voxel size = 1 mm × 1 mm × 1 mm, and slices = 180.

Preprocessing of resting-state fMRI data

Data were preprocessed by the Data Processing Assistant for Resting-State fMRI (DPARSF) version 4.0. (Chao-Gan & Yu-Feng, 2010). The first 10 volumes of each subject were deleted to allow the signal to reach equilibrium. The remaining 170 volumes were corrected for acquisition time delay between slices. All volumes were realigned to correct for head motion. Any subjects with head motion of >2.0 mm in maximum displacement or >2.0 rotation in angular motion were excluded. Individual functional images were normalized to the Montreal Neurological Institute (MNI) space using the Diffeomorphic Anatomical Registration Through Exponentiated Lie algebra (DARTEL) algorithm (Ashburner, 2007) and resampled into a voxel size of 3  mm × 3 mm × 3 mm. Cerebrospinal fluid signal, white matter signal, 24 head motion parameters and linear trend were regressed out as nuisance covariates.

For ReHo and DC, the time series of resting-state fMRI data of each voxel was linearly detrended and temporally bandpass filtered (0.01−0.08 Hz). The mean framewise displacement (FD) was calculated by averaging the FD of each subject across the time points. Each subject’s mean FD was included in all group-level analyses as a covariate to further control the head movement effect.

The analysis of dynamic ALFF, ReHo and DC

Temporal dynamic analysis (TDA) toolkits based on DPABI were used to analyze the dynamic regional indices (Yan et al., 2017). The sliding window-based method was adopted to explore the time-varying ALFF, dReHo or dDC in the entire brain (Chen et al., 2018a; Chen et al., 2018b; Chen et al., 2020a; Chen et al., 2020b). For each patient, the fMRI time series was segmented into 139 sliding windows with length of 64 s (32 TR) and sliding step of 2 s (one TR) to maximize statistical power within the window and for cross-level analyses (Allen et al., 2014). In each sliding window, ALFF, ReHo and DC were calculated. Regard to ALFF, the fast Fourier transform was computed based on the resting-state time series, and the cumulative power spectrum between 0.01 and 0.08 Hz in each voxel was estimated as the ALFF value of a given voxel. The ReHo value was analyzed by calculating Kendall’s coefficient of concordance of the time series of a given cluster to every 27 nearest neighboring voxels (Zang et al., 2004). For DC, functional connectivity in the whole brain between a given voxel and every other voxel was computed based on Pearson’s correlation coefficient (r) = 0.25 correlation thresholds in binary version (Yan et al., 2018). The standard deviation (SD) of ALFF, ReHo and DC values at each voxel were adopted to measure the variability of ALFF, ReHo and DC. The SD of ALFF, ReHo and DC were divided by the global mean dALFF, dReHo and dDC values to minimize the global effects of variability across subjects. Finally, an isotropic Gaussian kernel of four mm full-width-at-half-maximum (FWHM) was used to smooth the mean normalized dALFF, dReHo and dDC maps.

Statistical analysis

The normality of all demographic, clinical and fMRI variables was tested by using the Kolmogorov–Smirnov method and histogram inspection. For baseline data, nonparametric tests for data with a skewed distribution, Pearson’s chi-squared tests or Fisher’s exact tests for categorical data, and independent sample t-tests for data with a normal distribution were applied to assess group differences. The presented analyses were performed in the intention-to-treat (ITT) population, and missing data were mitigated by using the last observation carried forward method.

Functional MRI time series were modeled using a general linear model with SPM12 software (http://www.fil.ion.ucl.ac.uk/spm; Wellcome Trust Centre for Neuroimaging, London, UK) implemented in MATLAB 2017b (MathWorks, Natick, MA, USA). First, a whole-brain analysis was performed to compare the whole TR group sample with CR group subjects. A flexible factorial design modeling the factors subject, group, time and the interaction group × time was applied in second-level analysis (Kuncahyo et al., 2019). Differences in dynamic indicators between the TR group and CR group were investigated using a 2 × 3 flexible factorial design with the factor ‘group’ (two levers: TR, CR) and the factor ‘time’ (three levels: baseline, postrehabilitation, follow-up). In the current study, we aimed to investigate the differences in dynamic metrics between TR and CR groups over time; hence, we calculated the main effect of group and the interaction effect of group × time. A cluster defining threshold of p = 0.05 with familywise error (FWE) correction for multiple testing was used to assess for clusterwise significance. Statistical significance was determined using an extent threshold of 20 adjacent voxels.

The effect of the home-based TR approach on motor function was evaluated by using linear mixed-effects models of restricted maximum likelihood estimation within individuals. Time was analyzed as a repeated variable. Group effect, time effect and group-by-time interaction effect were inputted as fixed effects, and age, sex, educational level of patients, the number of treatment sessions, OT/PT and ETNS durations were entered as random effects. The changes in motor function scores from baseline to the end of the 12-week rehabilitative intervention and from baseline to the end of the follow-up period were analyzed. Partial correlation analyses were conducted in each group between the mean dALFF/dReHo/dDC value of each cluster, showing significant differences in the comparisons of the main effect of group or the interaction effect of group × time and motor function scores. Age, gender, educational level, the number of treatment sessions, OT/PT and ETNS durations were also considered as covariates, and the significance levels for correlation analyses were set at uncorrected p <0.05. A two-tailed p value of 0.05 was considered statistically significant for the analyses performed in SPSS version 24.0 statistical software (IBM Corporation, Armonk, NY, United States).

Results

Study population

The flowchart of the study is shown in Fig. 1. During the period from July 2017 to January 2019, 71 stroke patients were screened according to the inclusion/exclusion criteria. Fifty-two patients among these 71 were enrolled in the study, with 26 patients in each group, in accordance with the principle of random allocation. No participant has been excluded due to excessive head movement. Demographic characteristics (age, gender and educational level), clinical characteristics (high risk factors for stroke, time from stroke onset, hemisphere of infarction, lesion location, lesion volume, NIHSS and FMA) and the duration of the rehabilitation training (treatment sessions, duration of OT/PT and duration of ETNS) of the patients are shown in Table 1.

Figure 1 The flowchart of study.

Table 1 Demographic and clinical characteristics of patients groups.

Variables	Telerehabilitation group (n = 26)	Conventional rehabilitation group (n = 26)	P value	
Female	12(46.2)	14 (53.8)	0.579	
Age (y), means ± SD	64.19 ± 9.42	59.42 ± 10.00	0.083	
Education (y)	10 (7.0, 13,0)	10 (7.0, 12.3)	0.818	
High risk factors				
smoking	7 (26.9)	9 (34.6)	0.548	
hypertension	15 (57.7)	14 (53.8)	0.780	
Diabetes mellitus	15 (57.7)	14 (53.8)	0.780	
Dyslipidemia or obesity	10 (38.5)	7 (26.9)	0.375	
Atrial fibrillation	4 (15.4)	9 (34.6)	0.109	
TIA	1 (3.8)	2 (7.7)	0.548	
Time from stroke onset (d)	14 (13.0, 16.0)	14 (12.6, 16.0)	0.919	
Hemisphere of infarction			0.402	
Left-sided	10 (38.5)	13 (50.0)		
Right-sided	16 (61.5)	13 (50.0)		
Lesion location			0.674	
Basal ganglia	10 (38.5)	12 (46.2)		
Corona radiate	6 (23.1)	8 (30.8)		
Internal capsule	6 (11.5)	4 (15.4)		
Thalamus	4 (15.4)	2 (7.7)		
Lesion volume (ml)	4.2 (3.2, 5.4)	4.9 (3.4, 6.2)	0.176	
NIHSS	5 (3.0, 6.0)	5 (3.8, 8.0)	0.240	
FMA, means ± SD	71.88 ± 10.76	71.65 ± 10.25	0.937	
Treatment sessions	110.0 (96.0, 116.0)	97.0 (77.5, 108.0)	0.023	
Duration of OT/PT (h)	109.0 (95.7, 116.0)	97.0 (77.5, 108.0)	0.031	
Duration of ETNS (min)	2210.0 (1940.0, 2282.5)	1940.0 (1550.0, 2160.0)	0.019	
FD, means ± SD	0.15 ± 0.03	0.14 ± 0.02	0.273	
Notes.

Abbreviations SD standard deviations

TIA transient ischemic attack

NIHSS National Institutes of Health Stroke Scale

FMA Fugl-Meyer assessment

OT occupational therapy

PT physical therapy

ETNS electromyography-triggered neuromuscular stimulation

FD framewise displacement

Unless otherwise indicated, data are reported as medians with interquartile ranges in parenthesis or n(%).

Differences in dALFF

Main effect of group [SPM contrast: TR >CR and CR >TR] Compared with patients in the CR group, significantly increased dALFF variability in the right precuneus and bilateral precentral gyrus (PreCG) and decreased dALFF variability in the right inferior parietal lobule (IPL) were detected in patients in the TR group (Table 2 and Fig. 2A).

Table 2 Brain regions with significant differences in dynamical ALFF.

Comparisons	Brain regions/BA	Peak MNI coordinates	Cluster voxels	Peak t values	
		X	Y	Z			
Main effect of group: TR >CR	Precuneus, R/7	15	−45	48	31	6.64	
	Precentral Gyrus, L/4	−6	−33	72	22	6.52	
	Precentral Gyrus, R/4	8	−36	69	38	6.36	
Main effect of group: CR >TR	Inferior Parietal Lobule, R/40	51	−57	51	89	11.07	
Interaction effect: TR T2 >TR T1 and CR T1 >CR T2	Inferior Parietal Lobule, R/40	48	−57	51	95	12.15	
Interaction effect: CR T2 >CR T1 and TR T1 >TR T2	Cerebellum Anterior Lobe, R	−3	−54	−30	25	9.00	
	Rectal Gyrus, L/11	−3	15	−27	43	9.90	
Interaction effect: TR T3 >TR T1 and CR T1 >CR T3	Inferior Parietal Lobule, R/40	48	−57	51	123	12.43	
	Precentral Gyrus, R/4	42	−12	45	44	9.77	
	Middle Frontal Gyrus, R/9	42	18	39	54	9.42	
	Middle Temporal Gyrus, R/21	66	−36	0	21	7.65	
Interaction effect: CR T3 >CR T1 and TR T1 >TR T3	–	–	–	–	–	–	
Notes.

Abbreviations ALFF amplitude of low-frequency fluctuation

TR telerehabilitation

CR conventional rehabilitation

BA Brodmann’s area

MNI Montreal Neurological Institute

L left

R right

T1, at baseline; T2, at week 12; T3, at week 24.

Figure 2 Brain regions with significant differences in dynamic ALFF.

(A) Main effect of group [SPM contrast: TR & γτ, CR and CR & γτ , TR]; (B) interaction effect group × time I [SPM contrast: (TR T2 > TR T1 and CR T1& γτ; CR T2) ] and interaction effect group × time II [SPM contrast: (CR T2 > CR T1 and TR T1 & γτ; TR T2)]. (C) Interaction effect group × time III [SPM contrast: (TR T3 > TR T1 and CR T1 & γτ; CR T3)] and interaction effect group × time IV [SPM contrast: (CR T3 > CR T1 and TR T1 & γτ; TR T3)]. Family wise error rate corrected (p < 0.05, cluster size > 20 voxels). The color bar indicates the T value. ALFF, amplitude of low-frequency fluctuation; TR, telerehabilitation; CR, conventional rehabilitation; T1, at baseline; T2, at week 12; T3, at week 24.

Interaction effect group × time I [SPM contrast: (TR T2>TR T1 and CR T1>CR T2)] The interaction contrast analysis revealed significant dALFF changes in the right IPL (Table 2 and Fig. 2B).

Interaction effect group × time II [SPM contrast: (CR T2 >CR T1 and TR T1 >TR T2)] There were significant dALFF changes in the interaction contrast analysis in the right cerebellum anterior lobe and left rectal gyrus (RG) (Table 2 and Fig. 2B).

Interaction effect group × time III [SPM contrast: (TR T3>TR T1 and CR T1>CR T3)] The interaction contrast analysis revealed significant dALFF changes in the right IPL, right PreCG, right middle frontal gyrus (MFG) and right middle temporal gyrus (MTG) (Table 2 and Fig. 2C).

Interaction effect group × time IV [SPM contrast: (CR T3>CR T1 and TR T1>TR T3)] There were no significant results in the interaction contrast within the determined threshold (Table 2 and Fig. 2C).

Differences in dReHo

Main effect of group [SPM contrast: TR >CR and CR >TR] Compared with the dReHo variability of patients in the CR group, stroke patients in the TR group showed significantly decreased changes in the right fusiform gyrus (FG) (Table 3 and Fig. 3A).

Table 3 Brain regions with significant differences in dynamical ReHO.

Comparisons	Brain regions/BA	Peak MNI coordinates	Cluster voxels	Peak t values	
		X	Y	Z			
Main effect of group: TR >CR	–	–	–	–	–	–	
Main effect of group: CR >TR	Fusiform Gyrus, R/37	36	−75	−19	22	7.50	
Interaction effect: TR T2 >TR T1 and CR T1 >CR T2	Precuneus, R/7	12	−39	42	28	6.51	
Interaction effect: CR T2 >CR T1 and TR T1 >TR T2	–	–	–	–	–	–	
Interaction effect: TR T3 >TR T1 and CR T1 >CR T3	Middle Temporal Gyrus, L/13	−36	−75	15	27	9.72	
Interaction effect: CR T3 >CR T1 and TR T1 >TR T3	–	–	–	–	–	–	
Notes.

Abbreviations ReHo regional homogeneity

TR telerehabilitation

CR conventional rehabilitation

BA Brodmann’s area

MNI Montreal Neurological Institute

L left

R right

T1, at baseline; T2, at week 12; T3, at week 24.

Figure 3 Brain regions with significant differences in dynamic ReHO.

(A) Main effect of group [SPM contrast: TR & γτ, CR and CR & γτ , TR]; (B) interaction effect group × time I [SPM contrast: (TR T2 > TR T1 and CR T1& γτ; CR T2) ] and interaction effect group × time II [SPM contrast: (CR T2 > CR T1 and TR T1& γτ; TR T2)]. (C) Interaction effect group × time III [SPM contrast: (TR T3 > TR T1 and CR T1 & γ τ ; CR T3)] and interaction effect group × time IV [SPM contrast: (CR T3 > CR T1 and TR T1& γτ; TR T3)]. Family wise error rate corrected (p < 0.05, cluster size > 20 voxels). The color bar indicates the T value. ReHo, regional homogeneity; TR, telerehabilitation; CR, conventional rehabilitation; T1, at baseline; T2, at week 12; T3, at week 24.

Interaction effect group × time I [SPM contrast: (TR T2>TR T1 and CR T1>CR T2)]

The interaction contrast analysis detected significantly increased dReHo changes in the right precuneus (Table 3 and Fig. 3B).

Interaction effect group × time II [SPM contrast: (CR T2>CR T1 and TR T1>TR T2)] N o significant results were found in the interaction contrast analysis within the determined threshold (Table 3 and Fig. 3B).

Interaction effect group × time III [SPM contrast: (TR T3>TR T1 and CR T1>CR T3)] The interaction contrast revealed significantly increased dReHo changes in the left MTG (Table 3 and Fig. 3C).

Interaction effect group × time IV [SPM contrast: (CR T3>CR T1 and TR T1>TR T3)] There were no significant dReHo changes in the interaction contrast within the determined threshold (Table 3 and Fig. 3C).

Differences in dDC

Main effect of group [SPM contrast: TR >  CR and CR >  TR] Significantly increased dDC variability in the left cingulate gyrus (CG) was found in the TR group compared with the CR group (Table 4 and Fig. 4A).

Table 4 Brain regions with significant differences in dynamical DC.

Comparisons	Brain regions/BA	Peak MNI coordinates	Cluster voxels	Peak t values	
		X	Y	Z			
Main effect of group: TR >CR	Cingulate Gyrus, L/32	6	18	30	34	6.58	
Main effect of group: CR >TR	–	–	–	–	–	–	
Interaction effect: TR T2 >TR T1 and CR T1 >CR T2	Middle Frontal Gyrus, R/9	48	30	39	23	6.64	
Interaction effect: CR T2 >CR T1 and TR T1 >TR T2	–	–	–	–	–	–	
Interaction effect: TR T3 >TR T1 and CR T1 >CR T3	Precuneus, L/39	−36	−75	35	21	7.59	
Interaction effect: CR T3 >CR T1 and TR T1 >TR T3	–	–	–	–	–	–	
Notes.

Abbreviations DC degree centrality

TR telerehabilitation

CR conventional rehabilitation

BA Brodmann’s area

MNI Montreal Neurological Institute

L left

R right

T1, at baseline; T2, at week 12; T3, at week 24.

Figure 4 Brain regions with significant differences in dynamic DC.

(A) Main effect of group [SPM contrast: TR & γτ, CR and CR & γτ , TR]; (B) interaction effect group × time I [SPM contrast: (TR T2 > TR T1 and CR T1& γτ; CR T2) ] and interaction effect group × time II [SPM contrast: (CR T2 > CR T1 and TR T1& γτ; TR T2)]. (C) Interaction effect group × time III [SPM contrast: (TR T3 > TR T1 and CR T1& γτ; CR T3)] and interaction effect group × time IV [SPM contrast: (CR T3 > CR T1 and TR T1 & γ τ ; TR T3)]. Family wise error rate corrected (p < 0.05, cluster size > 20 voxels). The color bar indicates the T value. DC, degree centrality; TR, telerehabilitation; CR, conventional rehabilitation; T1, at baseline; T2, at week 12; T3, at week 24.

Interaction effect group × time I [SPM contrast: (TR T2 > TR T1 and CR T1 > CR T2)]

The interaction contrast analysis revealed significantly increased dDC alterations in the right MFG (Table 4 and Fig. 4B).

Interaction effect group × time II [SPM contrast: (CR T2>CR T1 and TR T1>TR T2)]

No significant dDC changes were detected in the interaction contrast within the determined threshold (Table 4 and Fig. 4B).

Interaction effect group × time III [SPM contrast: (TR T3>TR T1 and CR T1>CR T3)] Significantly increased dDC changes were found in the interaction contrast in the left precuneus (Table 4 and Fig. 4C).

Interaction effect group × time IV [SPM contrast: (CR T3>CR T1 and TR T1>TR T3)] No significant dDC changes were observed within the determined threshold (Table 4 and Fig. 4C).

Differences in motor function

The changes from baseline to the end of the 12-week rehabilitative intervention period and from baseline to the end of the 12-week follow-up for FMA scores are listed in Table 5. There were significant group-by-time interaction effects in FMA scores (p = 0.005) at week 12. This difference was not maintained after the 12-week follow-up period.

Table 5 Estimated mean change and effect sizes in FMA scores.

	TR (95% CI)	CR (95% CI)	P	Effect size	
			Group	Group * time	Time	MD	95% CI	Cohen’s d	
Change from the baseline to week 12 (end of rehabilitation)	11.14 (7.52, 14.71)	5.31 (3.45, 7.16)	0.181	0.005	<0.001	5.81	1.86, 9.75	0.82	
Change from the baseline to week 24 (end of followed-up)	13.01 (9.54, 16.61)	9.96 (6.93, 12.99)	0.842	0.099	<0.001	3.88	−0.75, 8.52	0.47	
Notes.

Abbreviations FMA Fugl-Meyer assessment

TR telerehabilitation

CR conventional rehabilitation

CI confidence interval

MD mean difference

Correlation between dynamic indices and motor function

There was a positive correlation between the increased dALFF variability in the left precentral gyrus and FMA changes from baseline to week 12 (r = 0.481, p = 0.032, uncorrected; Fig. 5) in the TR patients. Nevertheless, this difference was no significant after FWE correction. No other significant correlations between dReHo/dDC variability and FMA changes were observed in either the TR group or CR group.

Figure 5 The relationship between dynamic ALFF and FMA in the left precentral gyrus.

The dALFF variability at the left PreCG was positively correlated with the FMA scores of the stroke patients in TR group (r = 0.481, p = 0.032, uncorrected). dALFF, dynamic amplitude of low-frequency fluctuation; PreCG, precentral gyrus; TR, telerehabilitation; FMA, Fugl-Meyer assessment for upper and lower extremities.

Discussion

This reanalysis of a home-based TR training trial showed that the dynamic alteration patterns in regional intrinsic neural activity and degree centrality were different between the TR group and CR group. To the best of our knowledge, the current longitudinal study is the first to explore dynamic spontaneous neural activity and functional connectivity variability with TR training in stroke patients. Our study showed that dALFF alterations in the right precuneus, the bilateral PreCG, and the right IPL were significantly different between groups. The results revealed that the dALFF value in the right IPL was significantly increased within the TR group after 12 weeks of TR training. In contrast, significantly increased dALFF values in the right cerebellum anterior lobe and the left RG were observed within the CR group. After follow-up, we found that dALFF values in the right IPL, right PreCG, right MFG, and right MTG were significantly increased within the TR group relative to the CR group. Regarding dReHo, we found that dReHo values in the right precuneus and the left MTG significantly increased within the TR group after TR training and follow-up, respectively. In the analyses of dDC, brain regions of the left CG, the right MFG and the left precuneus showed significantly increased functional connectivity with the whole brain and the other brain regions in the TR group compared with the CR group. Furthermore, a relationship was observed between the altered dALFF variability in the left PreCG and FMA changes from baseline to week 12 in the TR group. These findings may provide some theoretical basis for the neural mechanism of TR training to improve the motor function of stroke patients and may extend our understanding of dynamic functional reorganization during the rehabilitation of stroke patients.

A previous study reported that the TR group showed significantly increased resting-state functional connectivity between bilateral PreCG areas at the end of rehabilitation (Chen et al., 2020a; Chen et al., 2020b). In line with the results of the previous study, we found that the spontaneous neural activity variability evaluated by dALFF in the PreCG areas was increased after TR training in the current study. The PreCG is a very important structure involved in executing voluntary motor movements and has a critical effect on executive and social function (Grefkes et al., 2008). A study has shown that functional or effective connectivity of the bilateral PreCG areas was increased among stroke patients with motor deficits, resulting from reduced transcallosal inhibition in stroke patients at acute or subacute stages (Cheng et al., 2020). Later, after stroke, when the patients had experienced some motor recovery, excessive activation in the PreCG was exhibited during paretic hand movement. During the chronic stage after stroke, the degree of functional recovery over time in patients was associated with reduced brain activation in the PreCG toward a more normal pattern (Green, 2003). Hence, it is speculated that the excessive regional intrinsic neural activity in the PreCG might occur as a compensatory mechanism, which might be regarded as a noticeable characteristic in the reconstruction and integration of the brain functional network. Based on the results of previous studies and this current study, we deduce that increased oxygen consumption and spontaneous neural activity might be conducive to accelerated functional reorganization after stroke. TR training as an effective rehabilitation method may be conducive to accelerating the process of normalization.

The PreCG is always the target brain location in noninvasive neuromodulatory techniques, such as rTMS and tDCS, which have potential for clinical utility in neurorehabilitation. After stroke, a series of functional reorganization changes occur that may result in reduced excitability in damaged brain regions and increased excitability in the homotopic regions of the undamaged contralateral hemisphere (Cicinelli et al., 2003). This disequilibrium between hemispheres is regarded as maladaptive and mainly regulated by abnormally increased interhemispheric inhibition from the contralateral to the ipsilateral hemisphere (Lee, Gunraj & Chen, 2007). Based on the above theories, the PreCG areas were targeted as the brain regions of interest in rTMS stroke rehabilitation for motor weakness aimed restoring this imbalance.

Similarly, tDCS modulated neuroplasticity by long-term potentiation and long-term depression resulting from synaptic strengthening or weakening to maintain the balance between hemispheres (Pulvermuller, 2018). Among the studies using tDCS, stroke patients received electric current stimulation either by ipsilesional anodal stimulation applied to the PreCG, or contralesional cathodal stimulation applied to the PreCG, or bilateral electrodes applied to PreCG areas (that is, ipsilesional PreCG anode, contralesional PreCG cathode) (Allida et al., 2020). Behavioral studies using tDCS have reported that either increasing the excitability of the ipsilesional PreCG or reducing the excitability of the contralesional PreCG improved motor function in stroke (Lindenberg et al., 2010; Hummel et al., 2005). A meta-analysis showed that anodal ipsilesional, cathodal contralesional and bilateral tDCS PreCG montages may play a part in motor recovery after subacute or chronic stroke, with moderate evidence (Liu et al., 2017b).

In this study, we demonstrated that the patients in the TR group showed significantly increased dALFF, dReHo and dD discrepancies in the precuneus, implying that the dynamic regional brain activity and functional connectivity in the precuneus were significantly heightened after TR training. The precuneus has extensive connectivity to cortical (the medial parietal cortex, the inferior and superior parietal lobules, the prefrontal cortex, the premotor area, the supplementary motor area, etc.) and subcortical (the thalamus, the dorsolateral caudate nucleus and putamen, etc.) structures to permit the brain to integrate both external and self-generated information and to regulate mental activity (Cavanna & Trimble, 2006). Unfortunately, we failed to discover a significant correlation between enhanced dynamic alterations in the precuneus and improvements in motor function. This implies that enhancement of spontaneous brain activity and functional connectivity in the precuneus may influence motor function by modulating cognitive function but does not act as a major factor during the improvement of motor function after stroke.

We also observed significant differences between the groups in the nonprimary sensorimotor network (SMN) areas, such as the MFG, MTG, IPS, RG and CG, which belong to the default mode network (DMN) and frontal-parietal network (FPN). Changes in neural activity of the DMN are likely to be related to enhanced neural function in cognitive and emotional control (Liu et al., 2017b). It has been reported that the DMN is involved in regulating cognitive function, such as episodic memory retrieval, visuospatial imagery, spatial navigation consciousness, processing emotionally salient stimuli, and coordinating the interactions of cognitive function and emotional processing (Zhang et al., 2017). The FPN involves executive skills and works as a hub for cognitive control for processing in contexts such as working memory, inhibitory control, the ability to shield from the influences of interfering information, and the capability to contrive and to adjust action plans (Stewart et al., 2016). These findings implied that neural functional reorganization might not only be confined to motor-related brain regions but may also involve brain areas associated with cognitive function and emotional regulation for patients with motor dysfunction in the process of rehabilitation after stroke.

In the current study, to explore whether the effects of TR on the increased dynamic dALFF variability in the left PreCG only occurred because participants in the TR group received more rehabilitation exposure, we evaluated the effects with variables for the number of treatment sessions and OT/PT and ETNS durations entered as covariates to control the potential confounding effects. The results showed that the differences in rehabilitation exposure had no effects on TR in the increased dynamic dALFF variability in the left PreCG for stroke patients, or that the effects were not large enough to be detected (for the change in increased dynamic dALFF variability in the left PreCG: the treatment sessions, F = 0.287, p = 0.597, 95% CI [−0.005–0.008]; the OT/PT durations, F = 0.238, p = 0.630, 95% CI [−0.005–0.008]; the ETNS durations, F = 0.511, p = 0.482, 95% CI [−0.002–0.004]). These unintended and individualized practices inherent in the TR approach, such as social participation activities and daily routines, together with the rehabilitation prescription, probably endow the TR program with the capability of providing more chances for stroke patients to experience and participate in realistic events and the social environment and might imperceptibly affect neural reorganization involving motor function and cognitive function.

Our study has several limitations. First, due to the small sample size of each group and the heterogeneity in hemispheres of infarction and lesion locations, it was difficult for us to further conduct subgroup analyses to determine the influences of locus of stroke on dynamic regional intrinsic neural activity and functional connectivity variability among patients in the TR group. Second, we aimed to explore the effects of the home-based TR approach on dynamic metrics and functional connectivity by calculating the main effect of group and the interaction of group × time; thus, we did not conduct analyses to assess the main effect of time. Third, because we assessed the regional spontaneous neural activity at the voxel level, it might be easier to detect the correlations between changed dynamic indicators and motor function if we utilized more refined assessments to evaluate motor function, such as grip strength, the action research arm test, the degree of spasticity and the range of motion of joints.

Conclusions

This study demonstrated that the home-based TR approach can alter the dynamic spontaneous neural activity patterns in multiple brain regions involved in the SMN, DMN and FPN in subcortical stroke patients with movement dysfunction. The significant association between dynamic intrinsic brain activity and improved motor function implied that neuroimaging indicators of dynamic regional spontaneous neural activity could be applied in studies to explore the mechanisms of brain reorganization in the process of rehabilitation in stroke patients with motor dysfunction. The identification of key brain regions promoting recovery would be conducive to providing a theoretical basis for developing noninvasive brain stimulation technology and providing strategies for formulating targeted rehabilitation programs.

Supplemental Information

Supplemental Information 1 The inclusion and exclusion criteria

Click here for additional data file.

Supplemental Information 2 Raw data

Click here for additional data file.

Supplemental Information 3 CONSORT Checklist

Click here for additional data file.

The authors thank the Institute of Science and Technology for Brain-Inspired Intelligence of Fudan University in Shanghai for valuable assistance.

Additional Information and Declarations

Competing Interests

Author Contributions

Human Ethics

Clinical Trial Ethics

Data Availability

Clinical Trial Registration

The authors declare there are no competing interests.

Jing Chen conceived and designed the experiments, performed the experiments, analyzed the data, prepared figures and/or tables, authored or reviewed drafts of the article, and approved the final draft.

Jing Li performed the experiments, analyzed the data, prepared figures and/or tables, authored or reviewed drafts of the article, and approved the final draft.

Fenglei Qiao performed the experiments, analyzed the data, prepared figures and/or tables, authored or reviewed drafts of the article, and approved the final draft.

Zhang Shi performed the experiments, prepared figures and/or tables, authored or reviewed drafts of the article, and approved the final draft.

Weiwei Lu performed the experiments, prepared figures and/or tables, authored or reviewed drafts of the article, and approved the final draft.

The following information was supplied relating to ethical approvals (i.e., approving body and any reference numbers):

The trial was approved by institutional review boards at the University of Ethics Committee of the Shanghai Fifth People’s Hospital of Fudan University (2014-ETRE-O66)

The following information was supplied relating to ethical approvals (i.e., approving body and any reference numbers):

The trial was approved by institutional review boards at the University of Ethics Committee of the Shanghai Fifth People’s Hospital of Fudan University (2014-ETRE-O66)

The following information was supplied regarding data availability:

The raw measurements are available in the Supplemental Files.

The following information was supplied regarding Clinical Trial registration:

ChiCTR-IPR-17011757

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
