# Peer review of "Effects of home-based telerehabilitation on dynamic alterations in regional intrinsic neural activity and degree centrality in stroke patients"

_PeerJ, doi:10.7717/peerj.15903_

## Round 0.1 · original submission · Major Revisions

All the reviewers raised major concerns. The authors should carefully address them.

·

Basic reporting

no comment

Experimental design

no comment

Validity of the findings

no comment

Additional comments

1. The title is too long to understand. I suggest the authors shorten it to make it simple and clear
2. In the abstract part, line 39, the authors said “…would provide a theoretical basis for noninvasive brain stimulation technology and strategies for formulating targeted rehabilitation programs”. In this sentence, provide a basis for what? No matter what diseases the patients suffer from? Please make it clear.
3. In the Introduction part, line 74, “…bilateral precentral gyrus (PreCG), also known as the primary motor cortex (M1),” as a well-known common sense, I suggest the authors delete this.
4. Why did the authors calculate both ALFF and ReHo in this study, as each can be used to measure the regional intrinsic activity?
5. The Introduction part is too long to read, please make it short and clear. Better around 650 words, no more than 700 words.
6. During the data processing process, were any subjects excluded due to excessive head movement? Please give more information about this.
7. Line 192, “3 mm ?3 mm ? 3 mm” What do these symbols mean?
8. Line 341, the authors said, “There was a significant positive correlation between the increased dALFF…” When the r-value is less than 0.5 and the p-value is not corrected, I do not consider this correction to be significant.
9. In the first paragraph of the discussion section, there are so many "after 12 weeks"
10. Line 405-417, Why did the author spend so much time discussing tDCS?
11. Line 419, what does the word “dD” mean?
12. Line 486, the authors said, “…motor function implied that neuroimaging biomarkers of dynamic regional…” How does the author define biomarker? Can these findings in this study really serve as biomarkers?
13. The language of this manuscript needs further polishing.

Reviewer 2 ·

Basic reporting

It this manuscript, Chen and colleagues describe interesting findings showing that home-based TR training can alter the patterns of dynamic spontaneous brain activity and functional connectivity in certain brain regions. In general, the paper is convincing and the results support the conclusions. I believe that it is an interesting paper, suitable for publication in Journal of PeerJ after major revisions as shown in below.

Experimental design

no comment

Validity of the findings

no comment

Additional comments

1. Firstly there needs to be some language editing for English expression - correct grammar and also correct wording. I have not commented on each occasion - most paragraphs (apart from the technical have errors).
2. In Table 1, there is statistically significant difference concerning the duration of the rehabilitation training (treatment sessions, duration of OT/PT and duration of ETNS) between the TR group and CR group. Does this overestimate the effect of TR.
3. Who measured your outcomes and were they blinded to group?
4. A limitation of the study is the cost-effectiveness aspect. In comparative trials of different service models this will be the deciding factor for funders. Please comment on this.
5. Your use of the Fugl Meyer is OK and justified and you correctly point out it is more sensitive but this is also its downfall - it is NOT a measure of meaningful functional recovery at an activity level. Besides, normally the FMA is not normally distributed -sorry if I missed this but did you transform otherwise it is usually necessary to report mean. And why CI and not sd for the change scores for FMA?

·

Basic reporting

No comment.

Experimental design

No comment.

Validity of the findings

No comment

Additional comments

Comments for authors:
This manuscript reports the results of a randomized controlled trial to determine effect of 12 week tele-rehabilitation (TR; versus conventional rehabilitation, CR) on motor function and structural/functional MRI measures, in 52 patients with acute (1-3 weeks post-ictus) subcortical stroke. The authors test the hypothesis that TR would manifest different dynamic change patterns in intrinsic brain activity and DC, and that the change neuroimaging indices would be be related to the improvement of motor function. Authors report improvement in motor scores, in dynamic ReHO and DC measures, and a large ALFF alteration, with modest relations between changes in dynamic ALFF and motor function. The manuscript should been major revised in the terms of following aspects:
1. The major concern is the apparent lack of effective randomization. Table 1 shows that participants in the CR group is somewhat younger, and most notably, received lesser number of sessions with shorter OT/PT and ETNS durations. Is it possible that TR appears to be as effective as CR only because participants in the TR group received more rehabilitation exposure? Authors only briefly mention this in the discussion, but this difference needs to be accounted for. Do the apparent effects remain if differences in rehabilitation exposure are accounted for in the statistical model?
2. The trial was not blinded (randomization was possibly blinded). Clearly the participants, therapists, and clinicians knew which treatment arm was assigned since one was an at home telerehabilitation while the other involved multiple in person trips to the clinic.
3. It appears no provision is made for missing data in the analysis since only the observed data is analyzed. This involves the restrictive assumption of missing at random which might be debatable (e.g. a patient getting worse might be inclined to drop out of the study). In addition, why were some patients dropped if they refused to do a repeat MRI ?
4. when you performed correlation analyses between the clinical measures and dALFF/dReHo/Ddc variability of each cluster showing significant group differences, you made it with TR patients and CR respectively or all participants together?
5. when you performed preprocessing, you already regressed 24 head motion parameters, why you made it again in your group-level analyses?
6. The authors should check carefully for the grammatical errors. The whole manuscript should be proof-read by a fluent English speaker.

---

## Round 0.2 · Major Revisions

There are still several concerns that should be addressed.

·

Basic reporting

none

Experimental design

none

Validity of the findings

none

Additional comments

1. In line 335, delete the word "significant", the r-value is less than 0.5 and the p-value is not corrected, this correction is not significant.
2. In line 480, I suggest the author change 'biomarker' to a more moderate tone word, as only imaging analysis results cannot be called 'biomarker'.

Reviewer 2 ·

Basic reporting

no comment

Experimental design

no comment

Validity of the findings

no comment

Additional comments

The authors have addressed all my questions I raised in my previous review.

·

Basic reporting

no comment.

Experimental design

no comment.

Validity of the findings

no comment.

Additional comments

The authors have provided responses to major concerns and revised the paper accordingly. Generally, the data seem reliable. However, there are still some minor issues that the author needs to further clarify.
INTRODUCTION
1. In the fifth paragraph of introduction section, the authors point out that “However, no studies have investigated the dynamic characteristics of spontaneous neural activity and functional connectivity in stroke patients with motor deficits during the process of rehabilitation” Do the authors confirm that there have been no studies using a similar approach in the stroke patients with motor deficits? Please consult some relevant literature again to clarify.
METHODS
2. In the methods section, the authors used 3mm×3mm×3 mm voxel size for the data resampling and an Gaussian kernel of 4 mm FWHM for smoothing. Is 4mm Gaussian kernel for smoothing appropriate? please clarify.

---

## Round 0.3 · accepted · Accept

This manuscript can be accepted now.

·

Basic reporting

no comment

Experimental design

no comment

Validity of the findings

no comment

Additional comments

no comment

·

Basic reporting

no comment

Experimental design

no comment

Validity of the findings

no comment

Additional comments

All queries I posed in my former review have been responded to by the authors.